# Effect of Bamboo Leaf Extract on Antioxidant Status and Cholesterol Metabolism in Broiler Chickens

**DOI:** 10.3390/ani9090699

**Published:** 2019-09-18

**Authors:** Mingming Shen, Zechen Xie, Minghui Jia, Anqi Li, Hongli Han, Tian Wang, Lili Zhang

**Affiliations:** College of Animal Science and Technology, Nanjing Agricultural University, No. 6, Tongwei Road, Xuanwu District, Nanjing 210095, China; 2017105056@njau.edu.cn (M.S.); 15117227@njau.edu.cn (Z.X.); jiaminghui3@163.com (M.J.); 15117409@njau.edu.cn (A.L.); hanhongli324@sina.com (H.H.); tianwangnjau@163.com (T.W.)

**Keywords:** antioxidant status, bamboo leaf extract, broiler, cholesterol metabolism

## Abstract

**Simple Summary:**

Cholesterol is an important lipid substance in organisms. As the precursor of bile acid, steroid hormones and vitamin D3, cholesterol plays important roles in lipid metabolism. Chicken is among the most consumed meat products worldwide; however, its cholesterol level is higher than that of other meat products. High cholesterol in a human diet will increase the risk of atherosclerosis. In addition, low-density lipoprotein cholesterol is susceptible to be oxidized, which will cause the death of broilers. Therefore, it is of great significance to enhance the antioxidant capacity and improve cholesterol metabolism in broiler chickens. Bamboo leaf extract (BLE) contains active ingredients such as flavonoids, polyphenols, and active polysaccharides, which possess anti-inflammatory, antioxidant and lipid-lowering effects. Our results show that supplementation of BLE in the basal diet improved growth and slaughter performance, antioxidant status and cholesterol metabolism in broilers. Therefore, the application of BLE as a feed additive has a certain economic value.

**Abstract:**

The objective of this study was to investigate the effects of dietary bamboo leaf extract (BLE) on antioxidant status and cholesterol metabolism in broilers. One-day-old male Arbor Acres (576) broilers were randomly divided into six groups. A control group was fed a basal diet, while five experimental groups were supplemented with 1.0, 2.0, 3.0, 4.0, and 5.0g BLE per kg feed in their basal diets. The result indicated that BLE supplementation linearly improved eviscerated yield and decreased abdominal fat (*p* < 0.05). A significant decrease of serum triglyceride (TG) and low-density lipoprotein cholesterol (LDL-c) content was observed with BLE supplementation (*p* < 0.05). BLE supplementation linearly improved the total antioxidant capacity and catalase activity in both serum and liver (*p* < 0.05). Glutathione peroxidase was quadratically increased in serum and linearly increased in the liver with BLE supplementation (*p* < 0.05). The malonaldehyde content in liver showed a linear and quadratic decrease with BLE supplementation (*p* < 0.05). BLE supplementation up-regulated the mRNA expression of cholesterol 7- alpha hydroxylase and low-density lipoprotein receptor and downregulated 3-hydroxy3-methyl glutamates coenzyme A reductase mRNA expression in the liver. The antioxidant enzyme mRNA expressions were all up-regulated by BLE supplementation in the liver. In conclusion, supplemental BLE improved antioxidant status and cholesterol metabolism in broilers, which eventually led to a decrease of serum TG, LDL-c content, and abdominal fat deposition.

## 1. Introduction

In the past few decades, the proportion of animal products in the human diet has increased considerably, and the high cholesterol content in animal products has attracted great interest of researchers [1,2]. If the human diet contains high concentrations of cholesterol from the animal products, the blood cholesterol concentration will raise, and the risk of hypercholesterolemia will increase [3,4]. As a result, atherosclerosis and coronary heart disease will happen because hypercholesterolemia is one of the major factors leading to these diseases [4]. As reported previously, every 1% reduction in serum cholesterol could reduce the incidence of coronary heart disease by 2% [5]. However, chicken products are widely consumed worldwide and are also the most cholesterol-containing meat [6]. Therefore, reducing the cholesterol content in broilers is of great importance for human health. In addition, there are many oxidative stresses in poultry production, such as heat stress, immune stress and transport stress; all these stresses will increase oxidizing substances in broilers [7]. If there is a large amount of low-density lipoprotein cholesterol (LDL-c) in chicken serum, LDL-c will react with oxidizing substance, and the oxidized LDL-c will be produced [8]. While the oxidized LDL-c is highly poisonous to the cells, it will cause damage to endothelial cells and accelerate the platelet adhesion and aggregation, release growth factors, cause hyperplasia of fibroblasts and organization, and eventually speed up the development of atherosclerosis [9,10]. As a result, it will cause sudden death of broilers and lead to economic losses. For all these reasons, it becomes a hot research topic and solutions need to be further explored for enhancing antioxidant status and improving cholesterol metabolism in poultry production.

Bamboo is widely distributed around the world, and its leaves have been used for medicinal and culinary purposes in China. It is reported that bamboo leaves contain active ingredients such as flavonoids, polyphenols, and active polysaccharides [11]. Research showed that bamboo leaf extract (BLE) has multiple biological effects, especially on cardiovascular and cerebrovascular protection. As mentioned before, BLE was able to reduce the cholesterol concentration of hyperlipidemia mice and improve liver function [12]. In addition to the effect of lowering serum cholesterol, BLE is capable to increase antioxidant capacity in hyperlipidemia mice [13]. Meanwhile, the research of Sunga et al. has shown that BLE reduced the adhesion of vascular epithelial factors, regulated endothelial cells to increase vascular mobility, and reduced the risk of atherosclerosis [14]. BLE has super antioxidant ability for scavenging free radicals in vitro [15,16], and improves antioxidant enzyme activity in vivo [17]. BLE is confirmed as an ideal choice for the body to supplement exogenous antioxidants and prevent hyperlipidemia, and some Chinese functional food or medicine made with BLE, like bamboo rice, bamboo beer, bamboo toothpaste, is very popular among Chinese consumers. The existing research about BLE in poultry production mainly focuses on the growth performance, meat quality and immune organ index [18]; little research has been conducted on the antioxidant effect. Furthermore, although multiple studies showed that flavonoids are able to affect fat metabolism in animals, and BLE holds the function to improve lipid metabolism in rat or mice, the effect of BLE on cholesterol metabolism in broiler chickens is still not clear. BLE supplementation in the basal diet could enhance antioxidant status and improve cholesterol metabolism in broiler chickens, which will be beneficial for poultry production and the human diet. Therefore, this study was conducted to investigate the effect of BLE on antioxidant status and cholesterol metabolism in broiler chickens.

## 2. Materials and Methods

### 2.1. Ethical Statement

Animal feeding experiments were carried out at Jiangpu farm of Zhujiang campus of Nanjing Agricultural University, and experimental analyses were conducted at the College of Animal Science and Technology of Weigang campus of Nanjing Agricultural University. All experimental procedures were approved by the Institutional Animal Care and Use Committee of Nanjing Agricultural University (GB14925, NJAU-CAST-2011-093), and the serial number of the laboratory animal use certificate issued by Science and Technology department of Jiangsu province is SYXK (Su) 2017–0007.

### 2.2. Animals, Diets and Experimental Design

Bamboo leaf extract (BLE) was obtained from Zhejiang XinHuang Biotechnology Co., Ltd. (Zhejiang, China), and its main components include flavonoids, polyphenols (the bamboo leaf flavonoid concentration is 70 mg per gram of BLE, and the polyphenol concentration is 50.42 mg per gram of BLE). A total of 576 one-day-old male Arbor Acres broiler chicks were obtained from a local commercial hatchery (Hewei Company, Anhui Province, China) and were randomly allotted into 6 groups with 6 replicates containing 16 birds each. Basal diets were designed for the starter phase (1–21 d) and growth phase (22–42 d) (Table 1). Chickens were supplied according to NRC (1994) recommendations for nutrition requirements. The control group (CON) was fed with a basal diet, while five experimental groups BLE1, BLE2, BLE3, BLE4, and BLE5 were fed the basal diet supplemented with 1.0, 2.0, 3.0, 4.0 and 5.0 g BLE per kg feed for 42 days. All birds were kept in three-layer pens; each replicate was divided into two pens. Temperature was maintained at 32–35 °C for the first five days, then gradually decreased to 22 °C and kept stable until the end of the experiment. During the trial period, birds had free access to feed and water. 

### 2.3. Slaughter Performance

Two birds from each replicate were weighted before slaughter and after sufficient exsanguination at 42 d. Then, the feathers, head, feet, abdominal fat and viscera were removed from the bird (except for kidney) and the carcass was reweighed. In addition, liver and abdominal fat weight were recorded separately. The right–side breast and thigh meat were weighed after removing skin and bone. The percentage weight of eviscerated yield, breast, thigh, abdominal fat and liver, compared with the live-weight was used to evaluate slaughter performance.

### 2.4. Sample Collection

At the end of the experiment, two birds (near the average body weight of each replicate) from each replicate were selected, and non-anticoagulant sterile blood vessels were used to collect blood samples from the jugular vein. The blood was left for 2 hours at 4 °C, then centrifuged at 3500 rpm for 10 min, and the supernatant was stored at −20 °C for analysis. A sample was cut from liver tissue from the middle of the left lobule in anatomical location and stored at −80 °C for antioxidant enzyme and mRNA expression analysis.

### 2.5. Cholesterol Metabolism Parameter Analysis

The total cholesterol (TC, kit number: A111-1-1), total triglyceride (TG, kit number: A110-1-1), high density lipoprotein cholesterol (HDL-c, kit number: A112-1-1), low density lipoprotein cholesterol (LDL-c, kit number: A113-1-1) and blood glucose (GLU, kit number: F006-1-1) in serum were measured by different commercial kits purchased from Nanjing Jiancheng Institute of Bioengineering (Nanjing, China). 

### 2.6. Serum and Liver Homogenate Antioxidant Enzyme Analysis

One gram of liver tissue from a sample preserved at -80°C was homogenized with 4.5 mL of 0.9% sodium chloride buffer with tube embed in ice by using an Ultra-Turrax homogenizer (Tekmar Co., Cincinnati, OH, USA), and the homogenates were centrifuged at 3500 rpm for 10 min. The supernatant was used to measure superoxide dismutase (SOD, kit number: A001-1-1), glutathione peroxidase (GSH-Px, kit number: A005-1-1), catalase (CAT, kit number: A007-1-1), total antioxidant capacity (T-AOC, kit number: A015-1-1) activities, and Malondialdehyde (MDA, kit number: A003-1-1) content by different commercial kits purchased from Nanjing Jiancheng Institute of Bioengineering (Nanjing, China) according to its instruction.

### 2.7. RNA Extraction and Quantitative Real-Time PCR

Trizol Reagent (Vazyme, NanJing, China) was used to extract total RNA from liver tissue, which was then treated by deoxyribonuclease I to remove the contaminant DNA. RNA was quantified based on the absorption of light by a Nanodrop ND-2000c spectrophotometer (Thermo Scientific, Camden, UK) at 260 nm (A260) and 280 nm. From each sample, 1 μg of RNA was used to synthesize cDNA in a 20 μL reaction mixture using the Primer-ScriptTM reagent kit (TaKaRa, Dalian, China) according to the manufacturers’ instructions. The real-time quantitative polymerase chain reaction was carried out by using the SYBR Premix Ex Taq II kit (TaKaRa) in an ABI 7300 fluorescence quantitative PCR instrument (Applied Biosystems, Foster City, CA, USA). The 20 μL reaction system included 10 μL of SYBR Premix Ex Taq buffer, 0.4 μL each of forward and reverse primers and dye, 2 μL of cDNA template, and 6.8 μL of distilled water. The real-time PCR cycling conditions were as follows: 95 °C for 30 s, 40 cycles of 95 °C for 5 s, and 60 °C for 30 s. The relative mRNA expression was determined using β-actin as an internal reference gene. The significance and correlation of quantitative results were analyzed by using 2‒ΔΔct as per Livak and Schmittgen [19]. Primer sequences are shown in Table 2.

### 2.8. Statistical Analysis

All data were preliminarily processed by using Excel 2016 and analyzed through one-way analysis of variance (ANOVA) using SPSS statistical software (Ver. 20.0 for Windows, SPSS, Inc., Chicago, IL, USA). The data were analyzed as a completely randomized design with a replicate as an experimental unit. Duncan’s multiple range test was performed to determine differences between treatments. The effect of BLE supplementation at various levels was evaluated using an orthogonal polynomial contrast test for linear and quadratic effects. Differences were regarded as significant at *p* < 0.05.

## 3. Results

### 3.1. Growth Performance 

In the starter phase, compared to the CON group, average daily feed intake was significantly higher in the BLE2 and BLE5 groups (*p* < 0.05), and average body weight in the BLE5 group increased significantly (*p* < 0.05). In the growth phase, compared with the CON group, average daily feed intake and average daily gain in the BLE2 and BLE5 groups were significantly higher than that of the CON group (*p* < 0.05). However, the BLE1 and BLE2 groups showed a significant decrease in feed: gain ratio (*p* < 0.05). Average daily gain and feed: gain ratio showed a quadratic improvement with increasing BLE dosage, and there was a linear and quadratic enhancement on average body weight when the BLE levels increased. During the whole rearing period, average daily gain and feed: gain ratio improved significantly in the BLE2 group over the CON group (*p* < 0.05). 

### 3.2. Slaughter Performance

As shown in Table 3, compared with the CON group, the percentage of eviscerated yield in BLE supplementation groups was significantly (*p* < 0.05) increased (except for birds in BLE3 group), the percentage of abdominal fat was decreased significantly in BLE supplementation groups (*p* < 0.05). In addition, the percentage of eviscerated yield showed a linear improvement (*p* = 0.007), and abdominal fat percentage showed a linear (*p* = 0.027) decrease with increasing level of BLE supplementation. Moreover, there was a quadratic decrease in the percentage of liver weight as BLE supplementation increased. No difference was observed in breast and thigh meat percentage among BLE groups (*p* > 0.05).

### 3.3. Serum Cholesterol Metabolism Parameters

Compared with the CON group (Table 4), the serum content of TG in BLE2, BLE3 and BLE4 groups were significantly decreased (*p* < 0.05), except for BLE1, the LDL-c content in serum was significantly decreased with increasing levels of BLE supplementation (*p* < 0.05). In addition, there was a quadratic (*p* = 0.002) decrease in the TG content and a linear (*p* <0.001) decrease in the LDL-c content with the increasing inclusion of BLE in the diet. No significant difference was observed in TC, HDL-C and GLU contents among groups (*p* > 0.05).

### 3.4. Antioxidant Index of Serum

Birds in BLE4 and BLE5 groups showed higher T-AOC activity in serum (Table 5) than other groups (*p* < 0.05). The CAT activity in serum of BLE2, BLE3 and BLE5 groups was significantly higher than that in the CON group (*p* < 0.05). Supplementation with BLE significantly increased GSH-Px activity in the serum of broilers (*p* < 0.05). In addition, BLE linearly increased T-AOC and CAT activity (*p* < 0.001, and *p* = 0.003), and quadratically increased GSH-Px activity in the serum as the addition level increased (*p* < 0.001), and GSH-Px activity in BLE2 and BLE3 groups was significantly higher than that in other BLE supplementation groups (*p* < 0.05). SOD activity and MDA concentration were not affected by BLE supplementation (*p* > 0.05). Birds in the BLE2 group showed a numerical minimum serum MDA concentration among groups. 

### 3.5. Antioxidant Index of Liver 

The effect of dietary BLE on the liver antioxidant index is shown in Table 6. Except for SOD in BLE4 group, the T-AOC and SOD activities in BLE supplementation groups were significantly higher than in the CON group (*p* < 0.05). Compared with the CON group, the CAT activity in BLE4 and BLE5 groups, and GSH-Px in the BLE5 group significantly improved (*p* < 0.05). In addition, linear (*p* = 0.028, and *p* <0.001) and quadratic (*p* = 0.009, and *p* = 0.007) increasing relationships between BLE level and T-AOC and CAT activities were observed, and there was a linear (*p* = 0.010, and *p* = 0.011) increase in SOD and GSH-Px activities as BLE supplementation increased, and SOD activity in BLE2 and BLE5 groups was significantly higher than in the BLE4 group (*p* < 0.05). Except for the BLE1 group, the MDA concentration was significantly decreased by BLE supplementation (*p* < 0.05); a linear (*p* = 0.014) and quadratic (*p* = 0.018) decrease effect was presented with increasing BLE. Moreover, a numerical minimum MDA concentration of liver was observed in the BLE2 group. 

### 3.6. Antioxidant Enzyme Gene Expression in the Liver 

As shown in Figure 1, the SOD, GSH-Px and CAT mRNA expressions were all up-regulated with BLE supplementation as compared to the CON group (*p* < 0.05). In addition, the GSH-Px mRNA expression in BLE5 was significantly higher than that in BLE1 and BLE2 groups (*p* < 0.05), and CAT mRNA expression in BLE3, BLE4 and BLE5 groups was significantly higher than that in BLE1 and BLE2 groups (*p* < 0.05).

### 3.7. Cholesterol Metabolism Related Gene Expression of Liver

Compared with the CON group, the cholesterol 7- alpha hydroxylase (CYP7A1) and low-density lipoprotein receptor (LDLR) mRNA expressions were significantly up-regulated by BLE supplementation (*p* < 0.05), and the CYP7A1 mRNA expression in BLE3 was significantly higher than in other BLE supplementation groups (*p* < 0.05). The highest LDLR mRNA expression was observed in the BLE4 group, and BLE2, BLE3 and BLE5 groups also showed favorable mRNA expression of LDLR compared with the BLE1 group (*p* < 0.05). In addition, the 3-hydroxy3-methyl glutamate coenzyme A reductase (HMGCR) mRNA expression was downregulated significantly except in the BLE1 group (Figure 2). Also, HMGCR mRNA expression in the BLE5 group was significantly lower than that in BLE2 and BLE3 groups (*p* < 0.05). 

## 4. Discussion

Muscle and fat are the main traits of carcass yield, while excessive fat deposition is a problem in the current poultry industry. It will not only affect broiler processing and feed conversion but also decrease carcass quality and the acceptance of consumers [20]. The present study indicated that BLE supplementation linearly improved eviscerated yield. Yang [21] reported that diet supplemented with 1.6 g bamboo leaf flavonoids per kg feed could improve carcass yield in broilers. As reported in the literature, flavonoids have a mild estrogen-like effect [22], which may contribute to muscle deposition of broilers. Bamboo leaf flavonoids are the major active components of BLE. It is reasonable to suggest that the linear improvement of eviscerated yield was attributed to increasing concentration of BLE. In addition, abdominal fat deposition was linearly decreased with BLE inclusion in the present study. It is well established that flavonoids regulate fat deposition and metabolism in animals. Li [23] reported that hawthorn leaf flavonoids reduced fat deposition in broilers in a dose-dependent manner. Most phytogenic flavonoids have a similar structure and functions [24]. Furthermore, Yang [25] demonstrated that BLE is confirmed to possess adipocyte differentiation properties. According to the present results, it is suggested that the flavonoids in BLE may contribute to reduce abdominal fat deposition. Although flavonoids in BLE play a major role in improving growth and fat metabolism, the lipid-lowering effect of polysaccharide in BLE cannot be ignored as it is reported that bamboo leaf polysaccharide could significantly decrease the liver fat content in mice [5], and the quadratic decline percentage of liver weight rate may result from the decrease of liver fat content in the present study.

Oxidative stress is a key factor leading to cardiovascular diseases, such as atherosclerosis, hyperlipidemia, inflammation and other chronic diseases [26]. When the redox state of the body is out of balance, the accumulation of reactive oxygen species will cause damage to vascular endothelial cells. On the other hand, the elevated TC and LDL-c in serum are the arch-criminal cause of cardiovascular diseases [27]. Thus, there are important links between antioxidant status, cholesterol metabolism and cardiovascular disease.

The antioxidant parameters in liver and serum indicate the antioxidant capacity of the organisms. Antioxidant enzymes like SOD, CAT and GSH-Px cooperate to eliminate excess free radicals and maintain homeostasis. The T-AOC refers to total antioxidant capacity, while MDA is one of the lipid peroxidation metabolites, reflecting the degree of oxidative stress. It is worth mentioning that CAT and GSH-Px activities in both serum and liver presented a dose-dependent enhancement as BLE increased. Zhang [28] reported that BLE could enhance GSH-Px enzyme activity in serum of aged rats. In addition, Zhang [29] demonstrated that liver-injured mice supplemented with bamboo leaf flavonoids showed improved GSH-Px enzyme activity and alleviated liver injury. As reported in the literature, the antioxidant effects of flavonoids in living systems are ascribed to their capacity to transfer electron free radicals, chelate metals catalysts, activate antioxidant enzymes, and inhibit oxidases [30]. It is reasonable to suspect that the flavonoids in BLE play a major part in improving antioxidant enzyme activities. However, the GSH-Px enzyme of serum in BLE2 and BLE3 groups was significantly higher than in other BLE groups. According to the research, BLE has peroxide scavenging capacity [31,32], so we speculated that with the increasing inclusion of BLE in the diet, the antioxidant capacity was mainly attributed to BLE itself. Our results showed that the MDA in liver tissue was linearly and quadratically decreased as BLE increased. Flavonoids and polyphenols play a role in inhibiting the formation of thiobarbituric acid-reactive substances [33]. In addition, studies showed that broilers supplemented with plant extracts rich in flavonoids, polyphenols and polysaccharide, such as *Ginkgo biloba* leave extract [34], and *Artemisia annua* extract [35], could enhance free radical scavenging capacity. In some animal models, like hyperlipemia [36], gastric mucosal damage [37] and myocardial ischemia reperfusion [38] rats, when supplemented with BLE remedy, the oxidative damage caused by these interventions was alleviated, accompanied by a low MDA concentration in serum or liver. According to our present results, it is suggested that BLE supplementation in broiler chickens improved antioxidant capacity and alleviate oxidative stress. Elisabeth [39] explained that a variety of electrophilic compounds including polyphenol and plant-derived constituents trigger the nuclear factor erythroid 2-related factor 2 pathway response. The SOD, CAT and GSH-Px are downstream genes of this pathway. Our results showed that mRNA expression in the liver was significantly up-regulated. It is reported that BLE can activate hepatic phase Ⅱ enzymes [40] or the AKT pathway [41] to improve inflammation and oxidative stress, all related to antioxidant effects. With the strictly demonstrated antioxidant effect of BLE in vitro studies [31,42], it is well-founded to speculate that BLE could both increase antioxidant enzymes and decrease chain breaking in broiler chickens, and these effects explain the linear improvement of T-AOC as BLE increased.

Cholesterol homeostasis is very important for broilers, as its metabolism dysfunction will lead to atherosclerosis, bile duct blockage or gallstones and other diseases [43,44]. The TC, TG, HDL-c, and LDL-c contents in serum are important indicators of cholesterol metabolism. In the present study, the TC and HDL-c concentrations in plasma were not affected by BLE supplementation. As reported in the literature, HDL-c is mainly secreted by the liver and small intestine, which plays a major role in transporting cholesterol and maintains a relative stable concentration itself [45]. Ding [12] and Liu [36] also found that BLE did not affect HDL-c concentration in hyperlipemia rats. These findings are similar to the present results. In addition, we speculated that the non-affected TC concentration with BLE supplementation in the present study may be attributed to the high content of TC in broilers [46]. The TG content in serum influences the accumulation of fat deposition. Our results present a quadratic decrease in TG and LDL-c contents in serum with BLE supplementation. According to the research, the TG, and LDL-c contents in serum were reduced significantly with BLE supplementation [13] in some hyperlipemia rat models. Furthermore, Yang [41] reported that BLE possesses an anti-inflammatory function in macrophages and inhibits adipogenic differentiation. A large number of studies have shown that flavonoids and polyphenols have regulatory effects on fat deposition and cholesterol metabolism in animals. Genistein [47], hawthorn leaf flavonoids [23], and soy isoflavone [48] reduce blood lipid and improve cholesterol metabolism in broilers. It is reasonable to speculate that bamboo leaf flavonoid possesses the same function, and supplementation of BLE could decrease the TG and LDL-c contents in serum of broilers and improve cholesterol metabolism. LDL-c is susceptible to free radicals, and high free radical scavenging capacity could reduce the oxidized LDL-c content [49]. It is speculated that the reduction of LDL-c concentration with dose may result from the antioxidant capacity improvement.

Although the lipid-lowering effects of BLE were strictly demonstrated, little research has been conducted on the effects of BLE on cholesterol metabolism and related mRNA expression in broiler chickens. The absorption, transformation, and synthesis of cholesterol metabolism are involved with some key regulators. For further investigating the effect of BLE on cholesterol metabolism, Quantitative Real-Time PCR was performed for these regulators. HMGCR is a rate-limiting enzyme in the whole process of cholesterol synthesis; increasing HMGCR activity will promote endogenous biosynthesis of cholesterol in the liver [50]. CYP7A1 catalyzes the conversion of cholesterol to bile acid; up-regulating CYP7A1 activity could decrease cholesterol levels [51]. LDLR mediates plasma LDL-c, which is ingested into cells for metabolism and degradation, and approximately 75% of plasma LDL-c was cleared in the liver [52]. Chen [47] reported that 50 mg/kg genistein significantly decreased HMGCR and CYP7A1 mRNA expression levels and increased LDLR mRNA expression in broiler liver. In addition, dietary of soy isoflavone showed the same outcomes in high-cholesterol diets rat [48]. Similarly, genistein exhibited the same effect of inhibiting HMGCR activity in cells [50]. Bamboo leaf flavonoid is the main compound of BLE, with a similar structure to genistein and soy isoflavone [30,53], and the present results are consistent with abovementioned studies in terms of HMGCR and LDLR mRNA expression. Furthermore, BLE showed an excellent inhibitory effect on HMGCR mRNA expression, indicating that BLE may reduce cholesterol synthesis and promote the clearance of LDL-c in serum. Studies on CYP7A1 activity regulation are not identical; both exogenous and endogenous cholesterol affect the expression of CYP7A1, and bile acids also have a negative feedback effect on CYP7A1. Our results showed significant up-regulated mRNA expression of CYP7A1, and higher CYP7A1 activity may contribute to the conversion of absorbed LDL-c to bile acid in the liver. By combining serum lipid parameters with the expression of liver cholesterol metabolism genes, it is deduced that BLE improved cholesterol metabolism by up-regulating LDLR and CYP7A1 mRNA expression to promote the conversion of LDL-c into bile acid, and down-regulated HMGCR expression to reduce cholesterol synthesis. However, due to the complicated mechanism of cholesterol metabolism, our study only presented a basic result of the effect of BLE on cholesterol metabolism, and the mechanism still needs to be further studied.

## 5. Conclusions

In conclusion, broiler chickens supplemented with BLE presented a linear improvement of eviscerated yield and reduced abdominal fat deposition. BLE supplementation improved antioxidant capacity by enhancing SOD, GSH-Px and CAT mRNA expression and reducing lipid oxidation, and a dosage of 2.0 to 3.0 g/kg presented the best outcome. Supplemental BLE decreased LDL-c concentration in serum, up-regulated mRNA expression of CYP7A1 and LDLR, and down-regulated mRNA expression of HGGCR. Supplementation of BLE improved cholesterol metabolism of broilers to some extent, but the specific mechanism needs further investigation.

## Figures and Tables

**Figure 1 animals-09-00699-f001:**
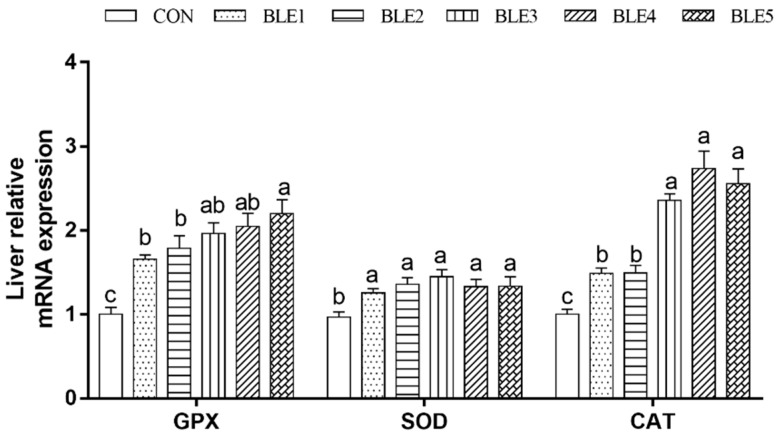
Effects of dietary BLE on antioxidant enzymes mRNA expression in liver of broilers. GSH-Px: Glutathione peroxidase; SOD: Superoxide dismutase; CAT: Catalase. Note: ^a.b.c^ means within the same gene of the histogram with no common superscript differ significantly (<0.05); CON: basal diet; BLE1, BLE2, BLE3, BLE4 and BLE 5 group, basal diet adding 1.0, 2.0, 3.0, 4.0 and 5.0g/kg BLE, respectively.

**Figure 2 animals-09-00699-f002:**
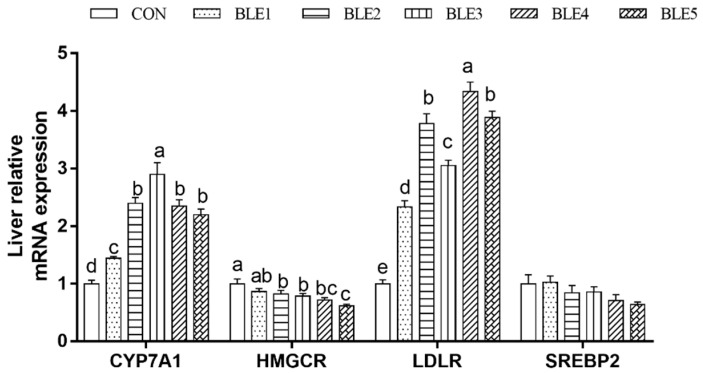
Effects of dietary BLE on cholesterol metabolism-related gene expression in liver of broilers. CYP7A1: cholesterol 7- alpha hydroxylase; HMGCR: 3-hydroxy3-methyl glutamates coenzyme A reductase; LDLR: low-density lipoprotein receptor; SREBP-2: sterol regulatory element binding transcription factor 2. Note: ^a.b.c.d^ means within the same gene of the histogram with no common superscript differ significantly (*p* < 0.05); CON: basal diet; BLE1, BLE2, BLE3, BLE4 and BLE 5 group, basal diet adding 1.0, 2.0, 3.0, 4.0 and 5.0g/kg BLE, respectively.

**Table 1 animals-09-00699-t001:** Composition and nutrient level of basal diets.

Item	Starter Phase (1–21 d)	Growth Phase (22–42 d)
Ingredient (%)		
Corn	57.02	61.36
Soybean	31.3	28.3
Corn gluten meal	3.7	1.7
Soya oil	3	4
Dicalcium phosphate	2	1.6
Limestone	1.2	1.3
L-Lysine	0.33	0.31
DL-Methionine	0.15	0.13
Sodium chloride	0.3	0.3
Premix ^1^	1	1
Nutrient levels ^2^		
ME (MJ/kg)	12.57	12.91
CP (%)	21.42	19.23
Lys (%)	1.20	1.10
Met (%)	0.50	0.44
Calcium (%)	1	0.93
Available Phosphorus (%)	0.46	0.39

^1^ Premix provided per kilogram of diet: VA 10 000 IU, VD_3_ 3 000 IU, VE 30 IU, VK_3_ 1.3 mg, thiamine 2.2 mg, riboflavin 8 mg, niacin 40 mg, choline chloride 600 mg, calcium pantothenate 10 mg, pyridoxine 4 mg, biotin 0.04 mg, folic acid 1 mg, VB_12_ 0.013 mg, zinc 65 mg, iron 80 mg, copper 8 mg, manganese 110 mg, iodine 1.1 mg, selenium 0.3 mg; ^2^ Calculated value.

**Table 2 animals-09-00699-t002:** Primer sequences used for Real-time PCR.

Gene Name ^1^		Primers Sequence (5′–3′)	Gene Bank Number
β-Actin	Forward	TGCTGTGTTCCCATCTATCG	NM_205518.1
	Reverse	TTGGTGACAATACCGTGTTCA	
CYP7A1	Forward	CACCATGGATCTGGGGAACA	NM_001001753.1
	Reverse	AGGCACATCCCAGGTATGGA	
LDLR	Forward	CTTCTGGTCTGACTGCGGTT	NM_204452.1
	Reverse	CAGAACACGGAGTCCTCGAA	
HMGCR	Forward	TTCTCGGCCGGGCGATTT	NM_204485.2
	Reverse	GGCACTCATAGTTCCAGCCAC	
SREBP-2	Forward	GTTCCTGGAGGTGTCAAGCA	AJ414379.1
	Reverse	CAGACTTGTGCATCTTGGCG	
SOD	Forward	CCGGCTTGTCTGATGGAGAT	NM_205064.1
	Reverse	TGCATCTTTTGGTCCACCGT	
CAT	Forward	GGTTCGGTGGGGTTGTCTTT	NM_001031215.2
	Reverse	CACCAGTGGTCAAGGCATCT	
GSH-Px	Forward	GACCAACCCGCAGTACATCA	NM_001277853.2
	Reverse	GAGGTGCGGGCTTTCCTTTA	

^1^ CYP7A1: cholesterol 7- alpha hydroxylase; LDLR: low-density lipoprotein receptor; HMGCR: 3-hydroxy 3-methyl glutamates coenzyme A reductase; SREBP-2: sterol regulatory element binding transcription factor 2; SOD: Superoxide dismutase; CAT: Catalase; GSH-Px: Glutathione peroxidase.

**Table 3 animals-09-00699-t003:** Effect of dietary BLE on slaughter performance of broilers.

Item: Percentage of (%)	Diet Treatment ^3^						SEM ^1^	*p* Value	
	CON	BLE1	BLE2	BLE3	BLE4	BLE5		Linear ^2^	Quadratic ^2^
Eviscerated yield	75.368 ^c^	78.233 ^a b^	77.862 ^a b^	76.665 ^b c^	77.618 ^a b^	78.475 ^a^	0.247	0.007	0.341
Breast meat	20.213	22.232	22.344	23.428	22.631	23.380	0.375	0.608	0.229
Thigh meat	18.763	18.154	18.119	18.206	18.332	18.088	0.208	0.533	0.610
Abdominal fat	1.278 ^a^	0.847 ^b^	0.900 ^b^	0.963 ^b^	0.902 ^b^	0.869 ^b^	0.042	0.027	0.085
Liver weight	2.536 ^a^	2.163 ^b^	2.113 ^b^	2.103 ^b^	2.145 ^b^	2.296 ^a b^	0.047	0.178	0.005

Note: ^a.b.c^ means within the same row with no common superscript differ significantly (*p* < 0.05); ^1^ standard error of the means; ^2^ Orthogonal polynomials were used to evaluate linear and quadratic responses to the levels of BLE treatment; ^3^ CON: basal diet, BLE1, BLE2, BLE3, BLE4 and BLE 5 group, basal diet adding 1.0, 2.0, 3.0, 4.0 and 5.0g/kg BLE, respectively.

**Table 4 animals-09-00699-t004:** Effect of dietary BLE on serum cholesterol metabolism parameters of broilers.

Item	Diet Treatment ^3^						SEM ^1^	*p* Value	
	CON	BLE1	BLE2	BLE3	BLE4	BLE5		Linear ^2^	Quadratic ^2^
TC (mmol/L)	4.478	4.417	4.104	4.244	4.293	4.287	0.226	0.367	0.235
TG (mmol/L)	0.404 ^a^	0.369 ^a b^	0.305 ^b^	0.327 ^b^	0.322 ^b^	0.373 ^a b^	0.010	0.148	0.002
HDL-c (mmol/L)	1.617	1.786	1.841	1.839	1.681	1.791	0.030	0.355	0.083
LDL-c (mmol/L)	2.958 ^a^	2.917 ^a^	2.488 ^b^	2.356 ^b^	2.401 ^b^	1.843 ^c^	0.068	<0.001	0.565
GLU (mmol/L)	12.523	13.080	12.444	12.358	11.931	12.713	0.124	0.299	0.473

Note: ^a.b.c^ means within the same row with no common superscript differ significantly (*p* < 0.05). TC: total cholesterol; TG: triglyceride; HDL-c: high-density lipoprotein cholesterol; LDL-c: low-density lipoprotein cholesterol; GLU: glucose; ^1^ standard error of the means; ^2^ Orthogonal polynomials were used to evaluate linear and quadratic responses to the levels of BLE treatment; ^3^ CON: basal diet, BLE1, BLE2, BLE3, BLE4 and BLE 5 group, basal diet adding 1.0, 2.0, 3.0, 4.0 and 5.0g/kg BLE, respectively.

**Table 5 animals-09-00699-t005:** Effect of dietary BLE on serum antioxidant index of broilers.

Item	Diet Treatment ^3^						SEM ^1^	*p* Value	
	CON	BLE1	BLE2	BLE3	BLE4	BLE5		Linear ^2^	Quadratic ^2^
T-AOC (U/ml)	5.848 ^c^	5.946 ^c^	6.804 ^b c^	6.778 ^b c^	8.099 ^a^	7.688 ^a b^	0.162	<0.001	0.818
CAT(U/ml)	5.405 ^b^	6.504 ^a b^	7.437 ^a^	7.919 ^a^	6.730 ^a b^	8.551 ^a^	0.288	0.003	0.302
SOD(U/ml)	162.928	167.121	163.238	165.775	168.467	164.067	1.027	0.565	0.476
GSH-Px(U/ml)	315.353 ^c^	350.001 ^b^	390.35 ^a^	383.333 ^a^	348.684 ^b^	345.175 ^b^	4.882	0.097	<0.001
MDA (nmol/ml)	3.098	3.013	2.842	3.077	3.184	3.034	0.056	0.713	0.535

Note: ^a.b.c^ means within the same row with no common superscript differ significantly (*p* < 0.05). T-AOC: total antioxidant capacity; CAT: catalase; SOD: Superoxide dismutase; GSH-Px: Glutathione peroxidase; MDA: malondialdehyde; ^1^ standard error of the means; ^2^ Orthogonal polynomials were used to evaluate linear and quadratic responses to the levels of BLE treatment; ^3^ CON: basal diet, BLE1, BLE2, BLE3, BLE4 and BLE 5 group, basal diet adding 1.0, 2.0, 3.0, 4.0 and 5.0g/kg BLE, respectively.

**Table 6 animals-09-00699-t006:** Effect of dietary BLE on liver antioxidant index of broilers.

Item	Diet Treatment ^3^						SEM ^1^	*p* Value	
	CON	BLE1	BLE2	BLE3	BLE4	BLE5		Linear ^2^	Quadratic ^2^
T-AOC (U/mg prot)	2.099 ^b^	2.479 ^a^	2.753 ^a^	2.527 ^a^	2.565 ^a^	2.545 ^a^	0.053	0.028	0.009
CAT (U/mg prot)	11.209 ^c d^	12.164 ^b c^	11.129 ^d^	12.089 ^b c d^	12.473 ^b^	13.976 ^a^	0.171	<0.001	0.007
SOD (U/mg prot)	516.916 ^c^	556.453 ^a b^	577.177 ^a^	555.953 ^a b^	537.516 ^b c^	583.752 ^a^	5.293	0.010	0.250
GSH-Px (U/mg prot)	63.602 ^b^	66.128 ^a b^	66.280 ^a b^	70.900 ^a b^	70.038 ^a b^	72.897 ^a^	1.190	0.011	0.928
MDA (nmol/mg prot)	1.733 ^a^	1.468 ^a b^	1.082 ^b^	1.214 ^b^	1.210 ^b^	1.266 ^b^	0.061	0.014	0.018

Note: ^a.b.c.d^ means within the same row with no common superscript differ significantly (*p* < 0.05). T-AOC: total antioxidant capacity; CAT: catalase; SOD: Superoxide dismutase; GSH-Px: Glutathione peroxidase; MDA: malondialdehyde; ^1^ standard error of the means; ^2^ Orthogonal polynomials were used to evaluate linear and quadratic responses to the levels of BLE treatment; ^3^ CON: basal diet, BLE1, BLE2, BLE3, BLE4 and BLE 5 group, basal diet adding 1.0, 2.0, 3.0, 4.0 and 5.0g/kg BLE, respectively.

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
