# Peer review of "Effect of Bamboo Leaf Extract on Antioxidant Status and Cholesterol Metabolism in Broiler Chickens"

_animals, 2019, doi:10.3390/ani9090699_

Round 1

Reviewer 1 Report

This manuscript reports beneficial effects og BLE in broilers, particularly regarding cholesterol metabolism and antioxidant capacity in serum and liver. Overall it is a good study and worth publishing. A few concerns remain, as listed below:

1. Language editing is required. Here are a few examples:

1a. " the proportion of animal food in human diet " - change "animal food" to animal product".

1b. Line 116 and line 117, extra text?

1c. "priority outcome" - change to "best outcome"?

1d. "The antioxidant parameters in liver and serum take present of redox state of the whole body" - this sentence is difficult to understand. 

2. Some abbreviations used in section 3.1 are difficult to follow.

3. Statistics in Tables 5 and 6 do not support such statments as" Birds in BLE2 group showed the lowest serum MDA concentration among groups", and " the lowest MDA concentration of liver was presented in BLE2 group".

4. " the quadratically decline percentage of liver weight rate may be resulted of the decrease of liver fat content in the present study." -  this is a very interesting speculation. If fat content can be measured in the liver samples, the additional results will reveal very important function of BLE. 

5. The "Conclusions" seem to be overconfident on the mechanisms behind the effect of BLE on cholesterol metabolism. Afterall, the authors only measure a few enzymes related to cholesterol metabolism on the RNA level, which doesnot provide enough support for confident conclusions.  

Reviewer 2 Report

Review

Journal: Animals (ISSN 2076-2615)

Manuscript ID: animals-580269

Type: Article

Title: Effect of bamboo leaf extract on antioxidant status and cholesterol metabolism in broiler chickens

General recommendations:

The present study focuses the effects of different amounts of bamboo leaf extract (BLE) supplementation in broiler chicken. Flavonoid containing products such as BLE are the focus of several researchers because of their anti-oxidative effects which plays an important role in the avoidance of oxidative stress in animals. Furthermore flavonoids are known to affect the fat metabolism in animals. The mentioned effects play a key role with respect to animal health and economic aspects. Therefore, the present study is from scientific interests. Although the implementation of the study sounds clear according to Material and Methods I have some critical points to the written manuscript.

The present introduction failed to introduce clearly the aim of the study and shows no relation to the discussion of the results. I missed a clear hypothesis and statements which explain why the authors conducted the present study. Did the authors think that BLE is helpful for the animal or for the human? Furthermore, I missed some adequate references in the introduction, especially in the first part of the introduction. Used references are not adequate to verify the given statements. The authors should explain all abbreviations before using in the main text. Differences in performance were reported but not shown in Table 3 as stated. In the discussion the authors cited the effects of flavonoids and emphasized the importance of flavonoids. Therefore it is not clear why the authors did not analyze these ingredients in the feed or BLE. Also in the discussion some statements have to prove by adequate references. In the subsection discussion, the authors have to discuss also the demonstrated differences between the different BLE groups otherwise the usage of different amounts makes no sense. The style of the references in the reference list has to be completely revised!

Line 10: Please capitalize cholesterol.

Line 23: Please capitalize „one“.

Line 25: g BLE per kg feed?

Line 40-41: Please add references

Line 43: Given reference do not verify the statement and the relationships between animal food cholesterol concentration and human blood cholesterol.

Line 44: Give references.

Line 45: add s to disease

Line 48 delete “than other meat products”

Line 47-54: Give adequate references. Reference 3 is not adequate to verify the given statement.

Line 54: “Accompanying the death…” It is only a sentence fragment

Line 77-79. Delete this statement because it is repeated in Materials and Methods. Perhaps give a statement about the institute or location were the study was conducted.

Line 82: Capitalize “diets”

Line 86- 87: delete, it is mentioned in the ethical statement.

Line 89: Because authors emphasize the content of flavonoids and polyphenols I missed data of concentrations of these components in the BLE.

Line 105: Describe blood sample collection.

Line 108: Please describe in detail the chosen location.

Line 110: Please change order of “slaughter performance” and “sample collection”.

Line 116-117: Subsection “Determination of Serum Biomarkers of Intestinal Permeability” is missing.

Line 121-129: Give kit numbers.

Line 158-165: What means CTR?

Line 158: ADFI was not found in Table 3. Also data of ABW, ADG and F/G are missed.

Line 165: To which data the statement “data not shown” refers?

Line 167, 182, 198, 215: Please capitalize “Table”.

Line 166: All measured traits of slaughter performance are given and should be given in g/ kg body weight.

Line 168: Replace “were” by “was” and “improved” by “increased”.

Line 171: What means positive effect?

Line 179, 195, 212, 232, 241, 255: What means CTR?

Line 234, 243: Please check capitalization.

Line 235: Please capitalize “figure”.

Line 235-236: Please also describe the significant differences between the BLE groups.

Line 247: Please write: “except for BLE1 group.”

Line 244-247: Please describe also significant differences between BLE groups.

Line 258: Please delete the first statement.

Line 263: Please check syntax.

Line 267: Please write: in the present study.

Line 265: Because the authors discuss the effects of flavonoids in detail, I missed the flavonoid analyses in the feed.

Line 314-315. Please add references.

Line 314-315: Is it important for human or for the animals/ broilers and why?

Line 341-342: Please add references.

Line 357: The authors concluded that BLE improved the cholesterol metabolism but the total cholesterol and HDL-c concentrations in plasma were not affected and the LDL-c concentration was affected dosage-dependent. This has to be discussed. A discussion focused only on gene expressions in the liver seemed to be an incomplete consideration of the present results.

Table 1: Please, delete the row “Total”.

Table 3: I think the data are given and should be given in g/ kg body weight. 

Round 2

Reviewer 2 Report

The manuscript was significantly improved by additional references and a comprehensible introduction. The conclusion also sounds clear. Here are some little suggestions before publishing.

Line 58: replace hot by important

Line 66: add Reference date

Line 76: add references

Please check spaces in the reference list.